# Difference between Keratinized- and Non-Keratinized-Originating Epithelium in the Process of Immune Escape of Oral Squamous Cell Carcinoma

**DOI:** 10.3390/ijms25073821

**Published:** 2024-03-29

**Authors:** Yoshiaki Kitsukawa, Chonji Fukumoto, Toshiki Hyodo, Yuske Komiyama, Ryo Shiraishi, Aya Koike, Shuma Yagisawa, Yosuke Kunitomi, Tomonori Hasegawa, Wataru Kotani, Kazuyuki Ishida, Takahiro Wakui, Hitoshi Kawamata

**Affiliations:** 1Department of Oral and Maxillofacial Surgery, Dokkyo Medical University School of Medicine, Shimo-Tsuga, Mibu 321-0293, Tochigi, Japan; y-kitsukawa@dokkyomed.ac.jp (Y.K.); chonji-f@dokkyomed.ac.jp (C.F.); hyodo14@dokkyomed.ac.jp (T.H.); y-komi@dokkyomed.ac.jp (Y.K.); ryo-s@dokkyomed.ac.jp (R.S.); a-koike362@dokkyomed.ac.jp (A.K.); s-yagi@dokkyomed.ac.jp (S.Y.); kunitomi@dokkyomed.ac.jp (Y.K.); hase-t@dokkyomed.ac.jp (T.H.); w.kotani@icloud.com (W.K.); 2Utsunomiya General Service Corps, Japan Ground Self-Defense Forces, Utsunomiya 321-0145, Tochigi, Japan; 3Section of Dentistry, Oral and Maxillofacial Surgery, Sano Kosei General Hospital, Sano 327-8511, Tochigi, Japan; 4Section of Dentistry, Oral and Maxillofacial Surgery, Kamitsuga General Hospital, Kanuma 322-8550, Tochigi, Japan; 5Concier Medical Lounge, Chiyoda, Tokyo 102-0074, Japan; 6Department of Diagnostic Pathology, Dokkyo Medical University School of Medicine, Shimo-Tsuga, Mibu 321-0293, Tochigi, Japan; ishida-k@dokkyomed.ac.jp

**Keywords:** immune escape, tumor immunity, immune checkpoint, oral squamous cell carcinoma, programmed death receptor-1 (PD-1), programmed cell death 1-ligand 1 (PD-L1), CD47

## Abstract

Immune checkpoint inhibitors (ICIs), including anti-programmed cell death 1 ligand 1 (PD-L1) antibodies, are significantly changing treatment strategies for human malignant diseases, including oral cancer. Cancer cells usually escape from the immune system and acquire proliferative capacity and invasive/metastatic potential. We have focused on the two immune checkpoints, PD-1/PD-L1 and CD47/SIRPα, in the tumor microenvironment of oral squamous cell carcinoma (OSCC), performed a retrospective analysis of the expression of seven immune-related factors (PD-L1, PD-1, CD4, CD8, CD47, CD56 and CD11c), and examined their correlation with clinicopathological status. As a result, there were no significant findings relating to seven immune-related factors and several clinicopathological statuses. However, the immune checkpoint-related factors (PD-1, PD-L1, CD47) were highly expressed in non-keratinized epithelium-originated tumors when compared to those in keratinized epithelium-originated tumors. It is of interest that immunoediting via immune checkpoint-related factors was facilitated in non-keratinized sites. Several researchers reported that the keratinization of oral mucosal epithelia affected the immune response, but our present finding is the first study to show a difference in tumor immunity in the originating epithelium of OSCC, keratinized or non-keratinized. Tumor immunity, an immune escape status of OSCC, might be different in the originating epithelium, keratinized or non-keratinized.

## 1. Introduction

Oral cancer is defined as a cancer occurring in the oral cavity, including the buccal mucosa, maxillary and mandibular gingiva, hard palate, tongue, mouth floor and lips, by the Union for International Cancer Control (UICC) [1,2,3,4]. Approximately 370,000 people per year worldwide are treated for oral cancer and more than 17,000 of these patients die of oral cancer [1,2,3]. The oral mucosa is covered by stratified squamous epithelia and is divided into keratinized epithelium and non-keratinized epithelium [5]. More than 90% of oral cancer are histopathologically diagnosed as squamous cell carcinoma (SCC) [6], but malignant diseases in the oral area include oral SCC (OSCC), salivary gland cancer, malignant melanoma, malignant lymphoma, sarcoma and metastatic cancer [7].

Surgery is the first choice for patients with resectable OSCC [7]. Postoperative radiotherapy or chemoradiotherapy is recommended for patients with adverse features found in a histological examination after surgery [7]. We routinely conduct postoperative treatment based on the evaluation of the risk of recurrence and metastasis of the tumor, and biological malignancy of cancer cells in patients with complete surgical resection [8,9]. We also perform appropriate surveillance after initial treatment, resulting in markedly higher survival rates in patients with OSCC [10]. However, some patients cannot undergo surgery due to locoregional tumor progression, distant metastasis, and tolerance of the patients for surgery. Moreover, locoregional recurrence and distant metastasis in patients who received the initial treatment may have difficulty with salvage treatment. These patients are to be treated by chemoradiation, while such patients with a history of radiotherapy and chemotherapy are to receive anti-programmed death 1 (PD-1) antibodies, nivolumab and pembrolizumab, according to the latest guidelines [7].

Immune checkpoint inhibitors (ICIs), including anti-PD-1 antibodies, are significantly changing treatment strategies for human malignant diseases, including oral cancer [11,12]. Cancer cells usually escape from the immune system and acquire proliferative capacity and invasive/metastatic potential (Figure 1). In the cancer immunoediting system, the PD-1/programmed cell death 1 ligand 1 (PD-L1) plays a central role [13]. PD-1 is the major receptor in immune cells as an activated checkpoint in the immune system and is expressed in T cells, B cells and natural killer cells activated in the tumor microenvironment [14,15,16]. PD-L1 is a PD-1 ligand that induces the phosphorylation of two motifs in PD-1. This interaction inhibits the proliferation and response of T cells; then, PD-L1 and PD-1 signaling is called the “Do not kill me signal” [16]. Another immune checkpoint molecule, Cytotoxic T-lymphocyte-associated protein 4 (CTLA-4), is also focused on the development of ICIs [7]. At present, seven ICIs involved in the PD-1/PD-L1 pathway and CTLA-4 signal are approved for clinical use [17], including the anti-PD-1 antibodies, nivolumab (anti-PD-1) [11,18], pembrolizumab (anti-PD-1) [12] and ipilimumab (anti-CTLA-4) [7]. Nivolumab and pembrolizumab can be used for head and neck cancer in Japan, but ipilimumab cannot be used for head and neck cancer in Japan. These antibody drugs have markedly changed the therapeutic strategy for unresectable head and neck cancer, and the expression level of PD-L1 in tumor cells and surrounding cells has been linked to the therapeutic effect of pembrolizumab [12]. Although the treatment with ICI is known to improve the prognosis in patients with recurrent and metastatic head and neck squamous cell carcinoma (HNSCC) including OSCC, the treatment efficacy of monotherapy remains limited [7,11,12,18,19]. Furthermore, neoadjuvant anti-PD-1 monotherapy has shown a relatively low major pathological response (MPR) rate (4.3% for pembrolizumab in HNSCC and 8% for nivolumab in OSCC) [20,21,22]. Therefore, several attempts are being made to block multiple immune checkpoints, but currently, only nivolumab in combination with ipilimumab has been used clinically in head and neck cancer, and is not highly effective [7,23].

CD47/signal-regulatory protein alpha (SIRPα) has been a recent focus as an immune checkpoint in macrophage/dendritic cells (DCs) [24,25,26]. SIRPα is an immunosuppressive receptor expressed in myeloid cells, including macrophages, DCs and neutrophils [25]. SIRPα is frequently overexpressed in cancer cells and binds to CD47, which is known to express on all normal cells [25,26]. The binding of CD47 and SIRPα causes SIRPα to activate multiple phosphatases, which results in the inhibition of the functions of SIRPα-expressing cells [25,26]. This signaling between CD47 and SIRPα is called a “Do not eat me signal”. An anti-CD47 antibody is now being tried against hematopoietic malignancies [27,28,29].

We have focused on the two immune checkpoints, PD-1/PD-L1, the “Do not kill me signal”, and CD47/SIRPα, the “Do not eat me signal”, to examine the kinetics of tumor immunity in the tumor microenvironment of OSCC (Figure 1). In this study, we performed a retrospective analysis of the expression of seven immune-related factors, including PD-L1, PD-1, CD47, CD4, CD8, CD56 and CD11c, and examined their correlation with clinicopathological status.

## 2. Results

### 2.1. Characteristics of Patients with OSCC

The study included 21 patients (13 male (61.9%), 8 female (38.1%)) with primary OSCC (Table 1. The mean age of the patients was 64.9 years and the median age was 66.5 years. The primary tumor sites were the tongue (n = 6, 28.6%), buccal mucosa (n = 3, 14.3%), mandibular gingiva (n = 8, 38.1%), and maxillary gingiva (n = 4, 19.0%). The T stage was classified as T1 (n = 2, 9.5%), T2 (n = 4, 19.0%), T4a (n = 14, 66.7%), and T4b (n = 1, 4.8%), and the N stage as N0 (n = 9, 42.9%), N1 (n = 5, 23.8%), N2a (n = 1, 4.8%), N2b (n = 4, 19.0%), and N2c (n = 2, 9.5%), resulting in an N-positive status in 12 patients (47.1%). Cancer stage was classified as Stage 1 (n = 2, 9.5%), 2 (n = 3, 14.3%), 3 (n = 1, 4.8%), and 4a (n = 15, 71.4%). Five patients (23.8%) died within 5 years after the first visit and three (14.3%) had recurrence or metastasis within 5 years.

### 2.2. Immunohistochemical Staining of Seven Immune-Related Factors

Results for the seven immune-related factors in the patients are shown in Table 2. The staining levels for PD-L1 were 0, +, ++ and +++ in 4, 12, 5 and 0 patients; 8, 4, 7 and 2 patients for PD-1; 4, 6, 4 and 7 patients for CD4; 3, 5, 7, and 6 patients for CD8; 0, 3, 6, and 12 patients for CD47; 16, 5, 0 and 0 patients for CD56; and 16, 5, 0 and 0 patients for CD11c. Based on the results, we divided the expression levels of the seven factors into two categories: positive (+) and negative (−), for statistical comparison (Appendix A).

### 2.3. Cancer Stage

#### Cancer Stage and Expression of Immune-Related Factors in OSCC

The expression status of the immune-related factors was compared with cancer stage (Figure 2). The stage showed no relationship with the expression of PD-L1 (*p* = 1.000), PD-1 (*p* = 0.119), CD47 (*p* = 0.338), CD11c (*p* = 0.278) or other factors.

### 2.4. Relationships among Expression Levels of Immune Checkpoint-Related Factors in OSCC

Correlations were examined among the immune checkpoint-related factors (PD-1, PD-L1, CD47 and CD11c) (Figure 3). There were no significant relationships between the expression levels of PD-1 and PD-L1 (*p* = 0.104), CD47 and CD11c (*p* = 0.338), and PD-L1 and CD11c (*p* = 0.272).

### 2.5. Comparison of the Expression Levels of Immune Checkpoint-Related Factors in Primary Tumor Sites (Keratinized or Non-Keratinized Epithelium)

Anatomical features and expression of immune checkpoint-related factors were compared by classifying primary sites into those with keratinized (maxillary and mandibular gingiva) and non-keratinized (tongue and buccal mucosa) epithelium (Figure 4). In all patients with tongue cancer, the tumor developed from non-keratinized mucosa in the ventral surface, rather than keratinized mucosa in the dorsal surface, then these were classified as the non-keratinized site origin. The primary sites in non-keratinized epithelium had significantly higher expression levels of CD47 (*p* = 0.001) and showed a tendency for higher levels of PD-L1 (*p* = 0.104) and PD-1 (*p* = 0.087). Double-positive cases for PD-L1/CD47 were significantly higher in the non-keratinized sites than those in keratinized sites (*p* < 0.001) (Figure 5). CD47(+)/CD11c(−) cases were also significantly higher in the non-keratinized sites (*p* = 0.001), and PD-L1(+)/PD-1(+) cases tended to be high in non-keratinized sites (*p* = 0.087) (Figure 5). There was no significance difference in the expression levels of PD-L1 (*p* = 0.676) and CD47 (*p* = 0.647) in normal epithelia at keratinized and non-keratinized sites on the same slides (Appendix A), indicating that the findings were tumor-specific.

## 3. Discussion

We examined the immune status of the OSCC tumor microenvironment by focusing on two immune checkpoints, PD-1/PD-L1, the “Do not kill me signal”, and CD47/CD11c, the “Do not eat me signal”. There were no significant findings relating to immune-related factors (PD-L1, PD-1, CD47, CD4, CD8, CD56 and CD11c) and several clinicopathological statuses. However, the immune checkpoint-related factors (PD-L1, PD-1, CD47) were highly expressed in non-keratinized epithelium-originated tumors when compared to those in keratinized epithelium-originated tumors.

In the oral mucosa, the attached gingiva, palatal mucosa and dorsal tongue are covered by keratinized squamous epithelium. In contrast, the floor of mouth, ventral surface of tongue, oral lip mucosa, buccal mucosa and soft palate are covered by non-keratinized squamous epithelium [5]. Keratinization in the epithelia is important for protecting the body from biological pathogens and extrinsic stress, i.e., mechanical (physical) and chemical stress [5]. It is unclear if both innate and adaptive immune mechanisms are activated, in addition to physical protection, in keratinized mucosa, or conversely, if these immune mechanisms are more activated in non-keratinized mucosa than keratinized mucosa. Immune cells may work differently in keratinized and non-keratinized sites. The buccal mucosa and ventral surface of tongue, which are non-keratinized epithelia, are common sites of oral lichen planus (OLP), a disease of T cell disorders [30]. It is of interest that immunoediting via immune checkpoint-related factors was facilitated in non-keratinized sites. Several researchers reported that the keratinization of oral mucosal epithelia affected the immune response [30,31,32,33]. However, our present finding is the first study to show a difference in tumor immunity in the originating epithelium of OSCC, keratinized or non-keratinized.

Blockade of the PD-1/PD-L1 axis is already used in clinical practice using several antibody drugs [11,12,16], and more than 100 clinical studies are ongoing [24]. Current clinical studies show a strong relationship between the PD-L1 expression level and clinical efficacy [12]. However, the response rate to PD-1/PD-L1 inhibition alone is not so high, and the antitumor activity of PD-1/PD-L1 inhibition alone is hardly shown in a “cold tumor” [24,25]. CD47, a cell surface molecule of the immunoglobulin superfamily, is another established target that is overexpressed in malignant cells [26,27,28,29,34,35]. CD47 binds to SIRPα, a receptor in various bone marrow-derived cells, and functions as an innate inhibitory checkpoint that inhibits phagocytosis against tumor cells and the activation of downstream responses [24,25,26,27,28,29,34,35]. When inhibiting the activation of innate immunity, tumor antigen presentation and priming of the T cell response, CD47 may enable tumor cells to evade both innate and adaptive immune surveillance [24,25,36]. Overexpression of CD47 has also been shown to function as a tolerance mechanism against PD-1/PD-L1 therapy [24,25,37]. Based on these mechanisms, the dual blockade of PD-1/PD-L1 and CD47/SIRPα immune checkpoint inhibitory signals has been examined [24,25].

This study has some limitations, such as a somewhat small-scale sample and a lack of information for the responses of tumors to ICIs. This study was initially conducted as a pilot study, which included all OSCC patients in one year without any exclusion criteria. Another limitation is the lack of examination for another major negative immune regulator, regulatory T cells (Treg). It was reported that Treg highly infiltrated in OSCC [38,39]. Treg and other factors that suppress tumor immunity, resulting in changes from hot to cold tumors, is a critical issue in immunotherapy. At present, it is difficult to show activated Treg by immunohistochemical staining alone. The examination of the existence of activated Tregs in OSCC might be necessary using fresh tissue. Furthermore, based on the current results, we are planning to examine the effects of the blockade of the PD-1/PD-L1 axis and/or CD47/SIRPα axis at keratinized and non-keratinized primary sites, and to evaluate the likely efficacy of the dual blockade of immune checkpoint inhibitory signals in non-keratinized mucosa.

## 4. Materials and Methods

### 4.1. Patients and Clinical Data

Twenty-one patients with OSCC who were treated in the section of oral and maxillofacial surgery at Dokkyo Medical University Hospital in 2014 and could be followed up for 5 years or more were included in this study. The patients who were treated with chemotherapeutic agents before having a biopsy or surgical materials were excluded. Clinicopathological data were obtained from electronic medical records. Age, gender, primary site, disease stage and prognosis were extracted as clinical information. Cancer staging was determined using the UICC TNM Classification of Malignant Tumours, 8th edition [4].

### 4.2. Immunohistochemical Staining

Immunohistochemical staining of specimens resected in biopsy or surgery was performed for PD-L1, PD-1, CD47, CD4, CD8, CD56 and CD11c (Table 3). Resected tissue was immediately fixed with 10% neutral-buffered formalin solution and paraffin-embedded solution to prepare 4 µm sections.

PD-1, CD4 and CD8 were stained with mouse anti-PD1 monoclonal antibody (clone. NAT105 ab52587, 1:50 dilution, Abcam, Cambridge, UK), mouse anti-CD4 monoclonal antibody (clone. 1F6 NCL-L-CD4-1F6, 1:40 dilution, Leica Biosystems, Wetzlar, Germany), and mouse anti-CD8 monoclonal antibody (clone. 4B11, R-T-U, PA0183, Leica Biosystems) using a BOND-III Fully Automated IHC and ISH Staining System (Leica Biosystems).

Other factors were stained as follows. The sections were deparaffinized with xylene and serially rehydrated with ethanol. Antigen was activated using a microwave at 95 °C for 10 min (pH 6.0 citrate buffer solution), washed with phosphate-buffered saline (PBS), and then treated with 0.3% hydrogen peroxide in methanol for 20 min for inhibition of endogenous peroxidase; this was carried out over a 30 min blocking time in total. X0909 Protein Block Serum-Free (Dako, Glostrup, Denmark) was used for blocking. Incubations with rabbit anti-PD-L1 monoclonal antibody (clone. 28-8 ab205921, 1:100 dilution, Abcam), rabbit anti-CD47 monoclonal antibody (clone. EPR21794 ab218810, 1:1000 dilution, Abcam), rabbit anti-CD11c monoclonal antibody (clone. D3V1E #45581, 1:400 dilution, Cell Signaling Technology, Danvers, MA, USA) and mouse anti-CD56 monoclonal antibody (clone. CD564 NCL-L-CD56-504, 1:100 dilution, Leica Biosystems) as primary antibodies were performed for 60 min. Thereafter, the procedure followed the polymer–immune complex method using EnVision (K4001, Dako) and EnVision FLEX (K8004, Dako). Staining intensity was graded using 2 to 4 stages per immune-related factor based on previous reports (Table 3). Based on the results, we divided the expression levels of the seven factors into two categories: positive (+) and negative (−), for statistical comparison (Table 3 and Appendix A). Examples of positively stained cells are shown in Figure 6.

### 4.3. Statistical Analysis

The staining status of immune-related factors was compared by clinical stage and characteristics of the primary sites of the tumors. Primary sites of the tumors were divided into those with keratinized squamous epithelium (maxillary and mandibular gingiva) and non-keratinized squamous epithelium (ventral surface of tongue, floor of mouth, and buccal mucosa). Data were evaluated by chi-square test or Fisher’s exact test (expected value ≤ 5) (two-sided 95% CI, *p* < 0.05) using IBM SPSS ver. 27. 0 (IBM Japan, Tokyo, Japan).

### 4.4. Ethical Standards

The study was approved by Dokkyo Medical University Hospital Ethics Committee (R-59-5-J). This was an opt-out study and no patients or their representatives requested exclusion from the study.

## 5. Conclusions

Tumor immunity, an immune escape status of OSCC, might be different in the originating epithelium, keratinized or non-keratinized.

## Figures and Tables

**Figure 1 ijms-25-03821-f001:**
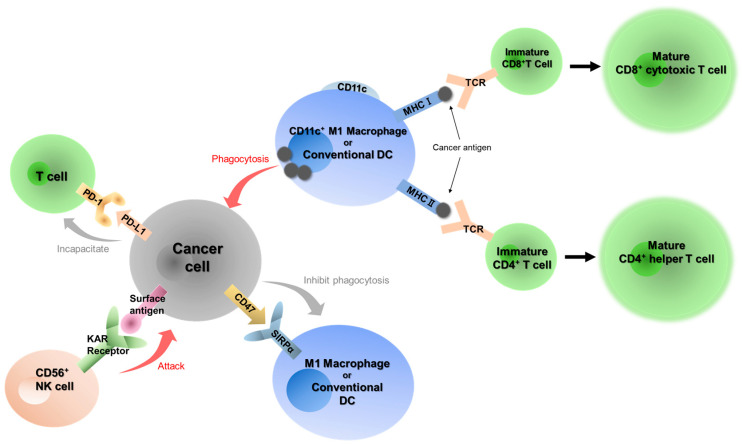
Immune response against cancer. Tumor cells develop an immune response via two immune checkpoints: the “Don’t kill me signal” through the PD-1/PD-L1 axis and the “Don’t eat me signal” through the CD47/SIRPα axis.

**Figure 2 ijms-25-03821-f002:**
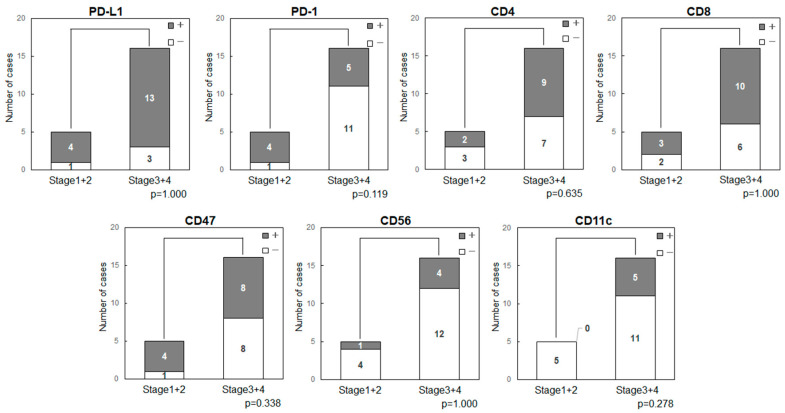
Comparison of cancer stage and expression status of immune-related factors in patients with OSCC. PD-L1 (*p* = 1.000), PD-1 (*p* = 0.119), CD47 (*p* = 0.338) and CD11c (*p* = 0.278) or other factors showed no significant relationships with the cancer stage.

**Figure 3 ijms-25-03821-f003:**
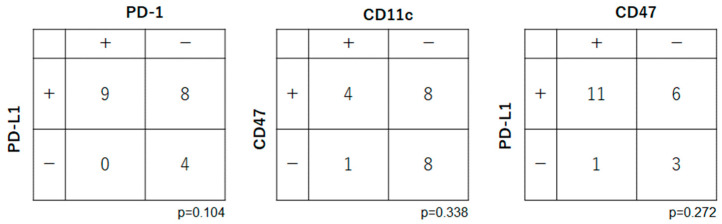
Expression of immune checkpoint-related factors in patients with OSCC. There were no significant relationships for PD-1 and PD-L1 (*p* = 0.104), CD47 and CD11c (*p* = 0.338) or PD-L1 and CD11c (*p* = 0.272).

**Figure 4 ijms-25-03821-f004:**
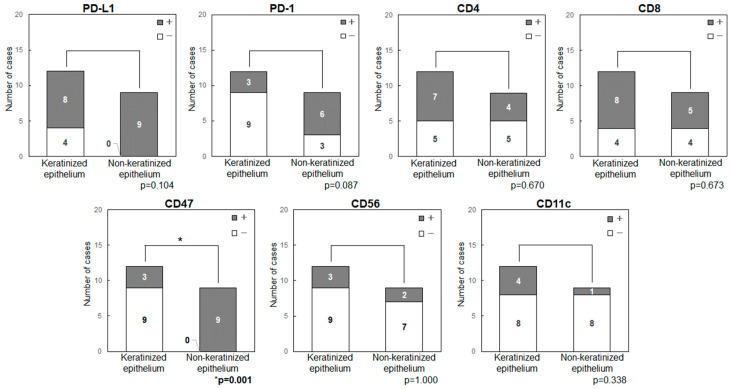
Expression of immune checkpoint-related factors in primary tumor sites (keratinized or non-keratinized epithelium) in patients with OSCC. The primary sites in non-keratinized epithelium had significantly higher expression levels of CD47 (*p* = 0.001) and showed a tendency for higher levels of PD-L1 (*p* = 0.104) and PD-1 (*p* = 0.087).

**Figure 5 ijms-25-03821-f005:**
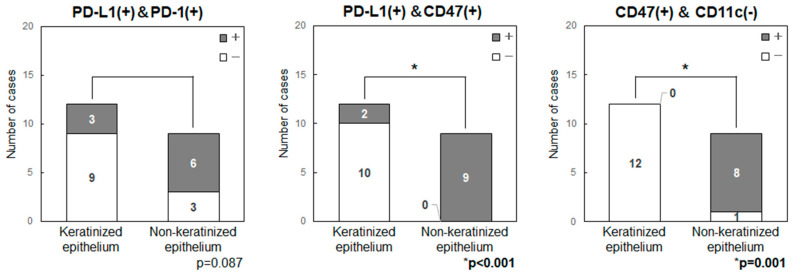
Combination of expression of immune checkpoint-related factors in primary tumor sites (keratinized or non-keratinized epithelium) in patients with OSCC. Double-positive cases for PD-L1/CD47 were significantly higher in the non-keratinized sites than those in keratinized sites (*p* < 0.001). CD47(+)/CD11c(−) cases were also significantly higher in the non-keratinized sites (*p* = 0.001), and PD-L1(+)/PD-1(+) cases tended to be high in non-keratinized sites (*p* = 0.087).

**Figure 6 ijms-25-03821-f006:**
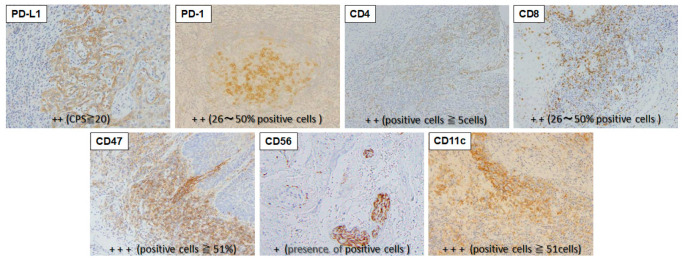
Examples of positive staining for each immune-related factor. The staining status of immune-related factors was assessed using the criteria in Table 3.

**Table 1 ijms-25-03821-t001:** Characteristics of patients with OSCC (n = 21).

Variable	n	%
Gender, male, n (%)	13	(61.9)
Gender, female, n (%)	8	(38.1)
Age, mean (SD) y	64.9	(15.4)
Age, median y	66.5	
Primary site, n (%)		
Tongue	6	(28.6)
Buccal mucosa	3	(14.3)
Mandibular gingiva	8	(38.1)
Maxillary gingiva	4	(19.0)
Clinical T stage, n (%)		
T1	2	(9.5)
T2	4	(19.0)
T4a	14	(66.7)
T4b	1	(4.8)
Clinical N stage, n (%)		
N0	9	(42.9)
N1	5	(23.8)
N2a	1	(4.8)
N2b	4	(19.0)
N2c	2	(9.5)
Clinical Stage, n (%)		
Stage 1	2	(9.5)
Stage 2	3	(14.3)
Stage 3	1	(4.8)
Stage 4a	15	(71.4)
Death in 5-year period, n (%)	5	(23.8)
Recurrence or metastasis in 5-year period, n (%)	3	(14.3)

**Table 2 ijms-25-03821-t002:** Expression of immune-related factors in each patient.

No.	Gender	Age	Primary Site	T	N	Stage	PD-L1	PD-1	CD4	CD8	CD47	CD56	CD11c
1	F	75	Maxillary gingiva	4a	1	4a	++	++	+++	+++	+++	+	+
2	M	65	Tongue	1	0	1	+	++	+	+	+++	ND	ND
3	M	61	Mandibular gingiva	4a	0	4a	+	+++	+++	+++	+++	ND	+
4	M	72	Mandibular gingiva	4a	1	4a	++	++	+++	+++	+	+	ND
5	F	75	Buccal mucosa	2	0	2	+	+++	+++	++	+++	+	ND
6	M	40	Tongue	2	0	2	++	++	+	++	+++	ND	ND
7	M	40	Maxillary gingiva	4b	2c	4a	+	ND	ND	ND	++	ND	ND
8	M	57	Tongue	4a	2a	4a	+	+	ND	+	+++	ND	ND
9	F	67	Mandibular gingiva	4a	0	4a	ND	+	+	++	+++	ND	+
10	F	72	Maxillary gingiva	4a	1	4a	ND	+	+	+++	+	ND	ND
11	M	71	Tongue	4a	2b	4a	+	ND	++	ND	+++	ND	+
12	M	72	Buccal mucosa	4a	2b	4a	+	ND	++	++	+++	ND	ND
13	M	41	Tongue	2	0	2	+	++	ND	++	+++	ND	ND
14	F	46	Maxillary gingiva	4a	2b	4a	++	ND	+++	++	+	+	+
15	M	73	Mandibular gingiva	4a	1	4a	++	ND	++	++	++	ND	ND
16	F	56	Tongue	2	1	3	+	++	++	+++	+++	+	ND
17	M	72	Buccal mucosa	4a	2b	4a	+	++	ND	+	+++	ND	ND
18	M	66	Mandibular gingiva	4a	2c	4a	+	ND	+	+	++	ND	ND
19	F	78	Mandibular gingiva	4a	0	4a	ND	ND	+++	+++	++	ND	ND
20	F	90	Mandibular gingiva	4a	0	4a	+	ND	+	+	++	ND	ND
21	M	75	Mandibular gingiva	1	0	1	ND	+	+++	ND	++	ND	ND

**Table 3 ijms-25-03821-t003:** Criteria for immunohistochemical staining of immune-related factors.

Immune-Related Factors	Summary of Factors	Criteria for Immunohistochemical Staining	Criteria for Immunohistochemical Staining (Two Classifications: Negative/Positive)
PD-L1	Immune escape from T cells in tumors.	ND: CPS < 1, +: 1 ≤ CPS < 20, ++: CPS ≥20	negative: CPS < 1, positive: 1 ≤ CPS
PD-1	Receptor on the surface of active T cells. Increased by T cell exhaustion.	Percentage of positive cells (×200) ND: <5%, +: 5–25%, ++: 26–50%, +++: ≥51%	negative: ND, +, positive: ++, +++
CD4	Marker of helper T cells.	Percentage of positive cells (×200) ND: ≤1 cell, +: 2–4 cells, ++: ≥5 cells, +++: cellular aggregate	negative: ND, +, positive: ++, +++
CD8	Marker of cytotoxic T cells.	Percentage of positive cells (×200) ND: <5%, +: 5–25%, ++: 26–50%, +++: ≥51%	negative: ND, +, positive: ++, +++
CD47	Immune escape from cDCs and M1 macrophages in tumors.	Percentage of positive cells (×200) ND: <5%, +: 5–25%, ++: 26–50%, +++: ≥51%	negative: ND, +, ++ positive: +++
CD56	Marker of NK cells.	Percentage of positive cells (×400) ND: negative, +: positive	same as on the left
CD11c	Markers of conventional dendritic cells (cDCs) and M1 macrophages.	Percentage of positive cells (×400) ND: <10 cells, +: 10–50 cells, ++: ≥51 cells	negative: ND, positive: +, ++

CPS: combined positive score, ND: not detectable.

## Data Availability

The data presented in this study are available on request from the corresponding author.

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
