# Peer review of "Difference between Keratinized- and Non-Keratinized-Originating Epithelium in the Process of Immune Escape of Oral Squamous Cell Carcinoma"

_ijms, 2024, doi:10.3390/ijms25073821_

Round 1
Reviewer 1 Report
Comments and Suggestions for Authors
I think is a good paper to be published
Author Response
Thank you.
Reviewer 2 Report
Comments and Suggestions for Authors
The article entitled with “An immune escape of oral squamous cell carcinoma might be different in the originating-epithelium, keratinized or non-keratinized” proposes a new perspective that immune escape status of OSCC might be different in the keratinized or non-keratinized originating-epithelium. This article is well organized and has potential clinical implications for the treatment of OSCC. However, I still have some concerns:
11. Expect for PD-1/PD-L1 immune checkpoints, the CTLA-4/B7-1/B7-2 are also classic ICIs and have also been use in clinical cancer therapy. Have authors analyzed these checkpoint proteins when they performed analysis of the expression of PD-L1/PD-1? If not, the authors should explain why they exclude the other classic ICIs in the introduction。
2. In figure 3, 5 and 6, the ordinate only shows the number of cases but not expression status of immune-related factors. The authors should add these specific expression values in the form of tables in the supplementary data.
3. In the cartoon pattern in Figure 1, the authors drew an angular tumor cell, which is not in line with the scientific setting. The authors should change the cell outline to a rounded line.
4. The discussion could include recent clinical evidence supporting the use of ICIs in OSCC management to add background information to the article.
5. The title of this article is exactly the same as the conclusions. In fact, this sentence is not suitable as a title. It is recommended that the title be changed to: Difference between keratinized or non-keratinized originating-epithelium in the process of immune escape of oral squamous cell carcinoma. Meanwhile, the authors can perform professional language polish to improve the readability.
Comments on the Quality of English Language
Just as comment 5 mentioned: The title of this article is exactly the same as the conclusions. In fact, this sentence is not suitable as a title. It is recommended that the title be changed to: Difference between keratinized or non-keratinized originating-epithelium in the process of immune escape of oral squamous cell carcinoma. Meanwhile, the authors can perform professional language polish to improve the readability.
Author Response
Q1: Expect for PD-1/PD-L1 immune checkpoints, the CTLA-4/B7-1/B7-2 are also classic ICIs and have also been use in clinical cancer therapy. Have authors analyzed these checkpoint proteins when they performed analysis of the expression of PD-L1/PD-1? If not, the authors should explain why they exclude the other classic ICIs in the introduction.
A: We did not analyze the CTLA-4/B7-1/B7-2 axis in this study. We know that a CTLA-4 inhibitor (ipilimumab) is clinically indicated for head and neck cancer in the United States, but its use is not approved in Japan. The efficacy of ipilimumab and nivolumab combination therapy is limited and the guideline recommendation level is not high in the US. So, we focused on the PD-1/PD-L1 axis, which is the main target of ICI in the current head and neck cancer therapy, and CD47/SIRPα axis as a new pathway that has not been applied clinically. In response to a suggestion, we added an explanation and references about it in the “Introduction” section.
Q2: In figure 3, 5 and 6, the ordinate only shows the number of cases but not expression status of immune-related factors. The authors should add these specific expression values in the form of tables in the supplementary data.
A: Figures 3, 5, and 6 simply show the IHC results for immune-related factors, divided as positive/negative by our criteria and the number of cases. Therefore, it is not possible to add specific expression levels of immune-related factors to these tables. Although the semi-quantitative evaluation of patient-specific immune-related factors has already been shown in Table 1, according to reviewer’s indication, the results were divided into positive/negative groups and prepared as separate graphs, which are attached as Supplementary Figure 1.
Q3: In the cartoon pattern in Figure 1, the authors drew an angular tumor cell, which is not in line with the scientific setting. The authors should change the cell outline to a rounded line.
A: As the reviewer pointed out, we corrected the shape of tumor cell in Figure 1.
Q4. The discussion could include recent clinical evidence supporting the use of ICIs in OSCC management to add background information to the article.
A: As pointed out, we added information about it (in not “Discussion”, but also the “introduction” section).
Q5: The title of this article is exactly the same as the conclusions. In fact, this sentence is not suitable as a title. It is recommended that the title be changed to: Difference between keratinized or non-keratinized originating-epithelium in the process of immune escape of oral squamous cell carcinoma. Meanwhile, the authors can perform professional language polish to improve the readability.
A: We changed the title to the one suggested by the reviewer. Please note that this manuscript has already been professionally proofread by a native English speaker prior to submission.
Reviewer 3 Report
Comments and Suggestions for Authors
This study evaluated the expression of immune related factors in OSCC. It is an interesting manuscript. In the introduction section the authors should add information regarding the clinical response rates of ICI in OSCC. The authors should further explain why only patients treated in 2014 were selected as a study group? Why not include larger sample from longer study period? Was this a prospective or a retrospective study? What were the exclusion criteria? The limitations of this study should be further explained in the discussion section.
Author Response
Q1: In the introduction section the authors should add information regarding the clinical response rates of ICI in OSCC.
A: As pointed out, we added information about it in the “introduction” section.
Q2: The authors should further explain why only patients treated in 2014 were selected as a study group? Why not include larger sample from longer study period? Was this a prospective or a retrospective study? What were the exclusion criteria? The limitations of this study should be further explained in the discussion section.
A: This is a retrospective study of patients treated at our institution in 2014 and we did not exclude any patients. The reason for the small sample size over a limited period is that we initially conducted this study as a pilot study. As pointed out, we added an explanation for it to the “Limitation” section.
Round 2
Reviewer 2 Report
Comments and Suggestions for Authors
Authors accurately addressed my concerns.